# Adaptation and Validation of the Program Sustainability Assessment Tool (PSAT) for Use in the Elementary School Setting

**DOI:** 10.3390/ijerph182111414

**Published:** 2021-10-29

**Authors:** Alix Hall, Adam Shoesmith, Rachel C. Shelton, Cassandra Lane, Luke Wolfenden, Nicole Nathan

**Affiliations:** 1School of Medicine and Public Health, College of Health, Medicine and Wellbeing, University of Newcastle, Callaghan, NSW 2308, Australia; Adam.Shoesmith@health.nsw.gov.au (A.S.); Cassandra.Lane1@health.nsw.gov.au (C.L.); Luke.Wolfenden@health.nsw.gov.au (L.W.); Nicole.Nathan@health.nsw.gov.au (N.N.); 2Hunter New England Population Health, Wallsend, NSW 2287, Australia; 3Hunter Medical Research Institute, New Lambton Heights, Newcastle, NSW 2305, Australia; 4Priority Research Centre for Health Behaviour, University of Newcastle, Callaghan, NSW 2308, Australia; 5Mailman School of Public Health, Columbia University, New York, NY 10032, USA; rs3108@cumc.columbia.edu

**Keywords:** sustainability, sustainment, maintenance, physical activity, measurement, development, reliability, school

## Abstract

There is a lack of valid and reliable measures of determinants of sustainability specific to public health interventions in the elementary school setting. This study aimed to adapt and evaluate the Program Sustainability Assessment Tool (PSAT) for use in this setting. An expert reference group adapted the PSAT to ensure face validity. Elementary school teachers participating in a multi-component implementation intervention to increase their scheduling of physical activity completed the adapted PSAT. Structural validity was assessed via confirmatory factor analysis. Convergent validity was assessed using linear mixed regression evaluating the associations between scheduling of physical activity and adapted PSAT scores. Cronbach’s alpha was used to evaluate internal consistency and intracluster correlation coefficients for interrater reliability. Floor and ceiling effects were also evaluated. Following adaptation and psychometric evaluation, the final measure contained 26 items. Domain Cronbach’s alpha ranged from 0.77 to 0.92. Only one domain illustrated acceptable interrater reliability. Evidence for structural validity was mixed and was lacking for convergent validity. There were no floor and ceiling effects. Efforts to adapt and validate the PSAT for the elementary school setting were mixed. Future work to develop and improve measures specific to public health program sustainment that are relevant and psychometrically robust for elementary school settings are needed.

## 1. Introduction

Promoting healthy behaviours during childhood is paramount to positive health and wellbeing [1,2]. Schools are an important setting for health promotion activities for children [3]. Numerous studies have investigated the effect of school-based public health interventions on student health outcomes, with systematic review evidence illustrating a positive impact on some outcomes [4,5,6,7,8,9]. In relation to specific physical activity programs, some reviews have found that formal policies and programs that focus on increasing and improving the quality of physical education [9,10,11], integrating short bouts of physical activity breaks throughout the day [10], provision of after-school physical activity programs [9,10], and promoting active transport to school [10,11] may promote physical activity [10,11] or physical fitness in children [9], although findings have been mixed. To optimise the benefits of school-based public health interventions, it is necessary that their delivery is continued over time, especially after external research support for the intervention has been withdrawn [12,13]. This is often referred to as intervention sustainment, which has been defined as the continued delivery of an intervention over time [14,15].

Sustainment of health-related interventions is a common challenge across a variety of settings and populations [12,13,16]. It has been found that once research support has been withdrawn, the full delivery of evidence-based interventions often declines or ceases [12,17,18,19]. For instance, in a recent review of the sustainment of school-based public health interventions, none of the 18 interventions assessed were sustained in their entirety, with at least one of the intervention components no longer delivered once external funding was withdrawn [18]. Failure to sustain delivery of public health interventions in schools results in a loss of population benefit and wasted resources. It may also impact communities’ trust in such programs and influence future uptake or participation [13].

An important part of continuing the delivery of school-based interventions is understanding and addressing the factors that impact intervention sustainment (i.e., determinants of intervention sustainment). It has been argued that multi-level determinants of intervention sustainment differ to some extent across diverse settings and populations [13], indicating the importance of understanding the specific determinants that may be salient within a particular setting (e.g., schools). Knowledge in this area is increasing with the publication of several recent reviews that aimed to identify barriers and facilitators to the sustainment of school-based public health interventions [18,20,21]. While these reviews provide important information on the possible determinants of the sustainment of public health interventions in schools, the factors identified are not always consistent or easily synthesised, due in part to the wide variation in the terminology, methods and measures used to classify determinants of sustainment in individual studies [20,21,22].

To this end, we must have consistent and comprehensive understanding of the determinants of sustainment. This requires the availability and use of valid, reliable and pragmatic measures that are appropriate and have been validated for the school setting. While there has been an increase in the number of theoretically informed and psychometrically evaluated measures of determinants of sustainment [23,24], few have been designed and evaluated for use specifically in the school environment. For example, in a recent review of implementation-related measures for school-based health policies, only three measures addressed sustainability [22].

In another review of implementation measures [24], only one of the sustainability measures was designed, developed and evaluated specifically for a school environment (the “School-Wide Universal Behavior Sustainability Index-School Teams” scale or SUBSIST). While the SUBSIST questionnaire is specific to the school context and has scored highly on psychometric and pragmatic rating scales in previous reviews [23,24], it was designed for a specific intervention [25] the “School-Wide Positive Behaviour Support” [25]. Consequently, SUBSIST may not be appropriate to identify factors impacting on the sustainment of school-based programs more broadly. If we are to build a comprehensive understanding of what factors need to be addressed in order to support the sustainment of school-based public health interventions, we must develop valid and reliable measures that are generalizable and relevant to a broad range of interventions. Such endeavours will also help to improve replication of findings, which is essential for building high-quality evidence.

There are, however, a number of other, more general measures of determinants of sustainment that may be appropriate for adaptation to the school setting [15,22,23,24]. In particular, the Program Sustainability Assessment Tool (PSAT), created by researchers at Washington University, is a 40-item tool that assesses the capacity of public health programs to be sustained across eight key domains, including: Environmental Support, Funding Stability, Partnerships, Organizational Capacity, Program Evaluation, Program Adaptation, Communications and Strategic Planning [26]. The measure has illustrated some evidence of reliability and validity in the context of chronic disease prevention programs [26]. Its development was informed by a sustainability-specific framework [27], which may reflect a range of important determinants of sustainment of public health interventions generally. Additionally, the measure is relatively short, easy to use [26] and is highly accessible and flexible with a license that allows for adaptations to be made [28]. Furthermore, the PSAT is targeted towards assessment of public health programs across a broad range of settings, including education [28]. While the current items and domains of the PSAT may require some amendments to allow for more specific assessment of sustainability determinants of public health programs in the elementary school setting, the generalised nature of the PSAT, with its solid theoretical underpinnings, strong pragmatic qualities and emerging psychometric properties, makes it a strong candidate for adaptation into a measure that is more specific to the school setting.

The overall objective of this study was to adapt and evaluate the relevance, reliability and validity of an adapted version of the PSAT to assess a physical activity policy within the elementary school setting. The specific aims were to:Adapt the PSAT domains and items to reflect the determinants of sustainment of a physical activity policy in the elementary school setting. In this specific study a physical activity policy was the target public health program, but with the view that the measure could be extended in future efforts to cover other public health programs in this setting.Evaluate the following psychometric properties of the adapted PSAT: structural validity, convergent validity, internal reliability, interrater reliability, floor and ceiling effects and norms.

## 2. Materials and Methods

### 2.1. Phase 1: Face and Content Validity

Content validity is the extent to which the items represent the constructs that a scale is designed to measure [29,30]. Face validity is a component of content validity and relates to the degree to which end-users determine the items as being an appropriate representation of the target constructs [29]. A group of experts and members of the target population, consisting of implementation scientists (*n* = 8), elementary school teachers (*n* = 4) and public health service delivery specialists (*n* = 4), reviewed and adapted the PSAT to ensure that the items adequately reflected the definitions of the domains, and were relevant and acceptable to the elementary school setting.

### 2.2. Phase 2: Psychometric Evaluation

Following adaptation of the PSAT, we undertook a formal psychometric evaluation of the adapted measure. The methods used are described below.

### 2.3. Design

The reliability and validity of the adapted PSAT were assessed as a secondary analysis using a convenience sample. Data were obtained from cross-sectional data collected from teachers following receipt of active implementation support from the intervention arms of two school-based cluster randomised controlled trials. Both trials assessed the impact of similar multi-strategy implementation strategies on increasing the mean minutes of physical activity scheduled by teachers across the school week. The first study was a pilot trial [31], with six schools randomised to receive the implementation strategies. Sustainment data collection occurred 18-months following active implementation support. The second study was a larger effectiveness trial [32], with 31 schools randomised to receive the implementation strategies. Sustainment data collection occurred six months after active implementation support ceased. Despite slight differences in the implementation strategies being assessed in the two trials, the physical activity intervention (i.e., scheduling of weekly classroom physical activity) that was the focus of the adapted PSAT was the same in both trials. Both trials received Human Research Ethics approval from Hunter New England (no. 06/07/26/4.04), the University of Newcastle (no. H-2008-0343) and relevant elementary school bodies.

### 2.4. Sample and Procedures

Elementary schools from the Hunter New England (HNE) region of New South Wales (NSW), Australia were eligible to participate in the trials if they were not participating in another physical activity intervention, and did not enrol only students who require specialist care [31,32]. HNE is a demographically and geographically diverse region in NSW Australia. It covers an area of approximately 130,000 km^2^, which includes densely populated regions such as metropolitan and regional hubs, as well as more geographically isolated areas such as rural and remote locations [33]. Socioeconomically, HNE is also very diverse, with areas of high wealth and other areas of poverty [33].

Following principal consent, schools were randomised to receive the multi-strategy implementation intervention or usual practice control. All classroom teachers from participating schools were invited to complete a self-report survey independently at three time-points (baseline, follow-up, and sustainment). Completion of the teacher survey was deemed consent to take part. Teachers from intervention schools completed a survey at all three time-points with the adapted PSAT items included in the sustainment data collection time-point. Only data from the sustainment data collection time-point from the intervention schools were included in this study. A complete description of the pilot [31] and effectiveness trials [32] are reported elsewhere.

### 2.5. Measures

#### 2.5.1. The Adapted Program Sustainability Assessment Tool (PSAT)

Based on stakeholder and end-user feedback (from Phase 1), an adapted version of the PSAT was evaluated. The adapted PSAT was reduced from 40 items across eight domains to 30 items across the following seven domains: Environmental Support, Funding Stability, Organizational Capacity, Program Evaluation, Program Adaptation, Communications and Strategic Planning. The Partnerships domain (5 items) was removed in addition to 11 items across the remaining domains (excluding Environmental Support), as these items were viewed as lacking relevance to this context. There were six additional items created relating to Organisational Capacity (e.g., resources and infrastructure, staff training); and Funding Stability (e.g., process for attending professional development), to ensure that constructs within these domains were adequately covered in relation to the school context. For each item respondents indicated the extent to which they agreed with the item, using a seven-point Likert scale, with response options ranging from “strongly disagree” to “strongly agree”. Consistent with the original measure, all items were positively worded with a higher level of agreement reflecting a greater capacity for program sustainment.

#### 2.5.2. Sociodemographic Characteristics

Teachers were asked to report their sex, year of birth, years of teaching experience, grade level taught and if they were a specialist physical education (PE) teacher.

#### 2.5.3. Sustainment of Scheduled Physical Activity at School

The implementation intervention was designed to increase teachers’ scheduling of physical activity across the school week. To assess this outcome teachers were asked to complete a daily activity logbook documenting the number of minutes they scheduled physical activity each day for one school week (5 days). The logbook included the time and occasion physical activity was scheduled for PE, sport or other structured activities (i.e., energisers and active lessons). This data provided an indication of teachers’ implementation of the current state Department of Education policy, which mandates that teachers schedule a minimum of 150 min of physical activity across the school week [34], and which was the target behaviour of the implementation intervention being tested in both trials. Sustainment of this behaviour was considered at 18 months following completion of the implementation intervention in the pilot trial [31] and 6 months in the effectiveness trial [32].

### 2.6. Statistical Analysis

Statistical analyses were undertaken in SAS version 9.3 and R version 4.0.2. The PSAT was developed based on a reflective measurement model, where it is theorised that the items in the scale are a manifestation of the same underlying construct, and thus are expected to be highly correlated and interchangeable [35]. Consequently, the psychometric indicators assessed in this study assume a reflective measurement model. Prior to assessing any psychometric properties, missing responses and response patterns were assessed for each item to identify any items that were poorly responded to and reviewed for possible exclusion. The polychoric correlations between all pairs of items were also calculated and reviewed to help identify any possible redundancies in the items. Items with a polychoric correlation coefficient above 0.8 [36] were reviewed for possible exclusion. A psychometric evaluation was then conducted on the resulting scale. An overview of the specific psychometric properties and the statistical analyses used are described below.

#### 2.6.1. Structural Validity

Structural validity is the extent to which the items in a scale are an adequate reflection of the hypothesized dimensionality of the construct being measured [35]. Structural validity is a component of construct validity [35], and is often assessed via factor analysis. As the dimensionality of the PSAT has been previously established and thus we have a clear hypothesis of how the items of the scale should relate to one another [37], a confirmatory factor analysis (CFA) proposing a seven-factor structure was conducted. To account for clustering of teachers within schools, we employed the procedures proposed by Hox [38] and outlined by Huang [39] to estimate a level one CFA model using the pooled within-cluster covariance matrix. This analysis provides an unbiased estimate of the model parameters by removing the group-level effects [39]. Pairwise deletion was used in the calculation of the covariance matrix, using all available data. Maximum likelihood was used as the estimation method. Parameter estimates were standardized with variances fixed at one [36]. An initial model assuming no correlation between factors was estimated and then revised to allow for such correlations, as it is reasonable to assume a relationship exists between the theoretical constructs. The following fit statistics and recommended criteria were used to assess the overall adequacy of the model:Standardized Root Square Residual (SRMR) < 0.08 [35,40];Comparative Fit Index (CFI) > 0.95 [35,40];Root Mean Square Error of Approximation (RMSEA) < 0.06 [35,40];Model Chi-Squared *p*-value > 0.05 [40].

To reduce selection bias we pre-specified the above fit indices, selecting those that were recommended as they have been found to be most insensitive to the sample size, model misspecification and parameter estimates used [40]. Modification indices and factor loadings were examined and used to revise the CFA model to ensure the most parsimonious, adequately fitting and theoretically justifiable model was selected. Specifically, items with low factor loadings (<0.40) or cross-loadings were examined for removal or model amendments. Standardized factor loadings and their associated standard error and *p*-value are reported.

#### 2.6.2. Convergent Validity Via Hypothesis Testing

Convergent validity is a component of construct validity of a self-report measure and involves assessing the relationship between the proposed scale and similar constructs [29]. The PSAT is designed to assess the capacity for program sustainability. Thus, if the adapted PSAT accurately reflects its intended construct, i.e., capacity for program sustainability, adapted PSAT scores should be positively related to sustainment of the target behaviour, as has been qualitatively observed to some extent in a previous study using the original PSAT measure [41]. Based on the theoretical construct that the PSAT is intended to measure and findings from previous research, we hypothesised that the seven domains of the adapted PSAT as well as the total adapted PSAT score would be positively related to the teacher’s total minutes of scheduled physical activity across the school week at the sustainment data collection phase, as this was the target behaviour that we were attempting to sustain in the two intervention trials.

To assess this hypothesis, adapted PSAT domain scores were calculated by summing together each item in a domain and dividing by the number of non-missing items. Scores were only calculated for teachers who answered a minimum of 50% of items from each domain. Linear mixed regression models were used to assess the relationship between the seven adapted PSAT domains and total adapted PSAT score with teachers’ total minutes of scheduled physical activity at the sustainment phase of data collection. A separate model was conducted for each domain and the total score, with a random level intercept for school to account for clustering. A positive relationship between the adapted PSAT scores and the number of minutes teachers scheduled physical activity (i.e., intervention target) was hypothesised.

#### 2.6.3. Internal Reliability

Internal reliability is the extent to which items in a domain or scale are correlated [30]. Cronbach’s alpha were calculated for each domain, with values between 0.7 and 0.95 considered acceptable [30].

#### 2.6.4. Interrater Reliability

To assess the degree to which teachers from the same school were reporting similar adapted PSAT scores, we calculated the intraclass correlation coefficients (ICC), ICC(1) and ICC(2), for each domain as well as for the overall adapted PSAT score using a linear mixed effects model. The ICC(1) presents the proportion of variance in the adapted PSAT scores that is attributable to school membership (i.e., between-group) [42]. ICC(2) represents the reliability of the group-adapted PSAT scores [42]. Similar to other psychometric evaluations of implementation measures, a threshold of 0.70 for ICC(2) was used to indicate adequate group level reliability [43].

#### 2.6.5. Floor and Ceiling Effects

Potential responsiveness was evaluated by assessing absence of floor and ceiling effects. Scales that illustrate limited floor and ceiling effects have an ability to capture future changes in the construct being measured. The percentage of respondents reporting the lowest and highest possible score for each domain were calculated. Domains where >15% of respondents obtained the lowest (floor) or highest (ceiling) score were considered indicative of floor and ceiling effects [30].

#### 2.6.6. Norms

The mean, standard deviation, median, minimum and maximum for each of the adapted PSAT domains and total adapted PSAT score were calculated.

## 3. Results

### 3.1. Phase 1: Face and Content Validity

Substantial adaptation was required to ensure relevance and face validity of the PSAT for the elementary school setting. Each domain was assessed to ensure its relevance to the school setting with one of the original eight PSAT domains removed: the Partnership domain (5 items). As many health programs and policies relevant to schools are often determined by educational departments or boards, the establishment of partnership with stakeholders was considered outside the scope of individual schools, particularly teachers, and more relevant to the governing bodies of schools. The remaining seven domains (originally consisting of 35 items) were kept and the items within each reviewed and assessed for relevance, appropriateness and acceptability to the school setting. As a result of this process, 11 items from the original PSAT were removed: three from Funding Stability; two from each of the Strategy Planning, Program Adaptation and Communications domains; and one each from the Organisational Capacity and Program Evaluation domains. Most of the items removed were perceived by stakeholders and experts as not relevant to individual schools and more relevant to a school’s governing body (e.g., a department of education, school board) (see Appendix A). Six additional items were created as they were identified as potentially important determinants of program sustainment in the school setting by the expert group. Specifically, four were added to the Organisational Capacity domain and two to the Funding Stability domain (see Appendix A). The 24 remaining original items from the PSAT were reworded to ensure they were relevant and specific to the school setting. For example, wording changes consisted of swapping “the program” as the subject to “my school”, as it did not make sense that the program performed the actions in most of the items. We also included the specific program of interest, in our case the scheduling of physical activity, but with a view that this could be changed depending on the specific program being evaluated. This was done to improve clarity and ensure that respondents were interpreting all items consistently and with reference to the program of interest. Finally, we also added in an example to some of the items, to again improve clarity and consistency of reporting (see Appendix A for comparison of the original PSAT items with the amended items). From this process, a total of 30 items across seven domains were included in the teacher survey as part of the initial adapted PSAT and included in the psychometric evaluation.

### 3.2. Phase 2: Psychometric Evaluation

Two hundred and sixty-one teachers from 33 (89%) schools returned a survey, of which 45 (17%) missed all 30 items and two missed more than half of all items. This left a sample of 214 participants from 30 schools who were included in the quantitative analyses as they answered a minimum of 50% of the adapted PSAT items; 187 participants answered all adapted PSAT items. School and participant characteristics are outlined in Table 1.

### 3.3. Item Assessment

Missing values were low for all 30 items, ranging from 0.47% to 4.70% (see Table 2). The full range of response options were used for 27 of the 30 items, although a left-hand skew was observed for all 30 items, with less than 3% of respondents utilising the lower end of the response scale, and most participants answering towards the positive end of the scale. For three items from the Organisational Capacity domain, the “strongly disagree” response option was not endorsed by any of the respondents. These items included: “My school has enough trained school champions to support the scheduling of physical activity,” “School champions and teachers at my school have enough supervision and support to implement the scheduling of physical activity” and “The level of school champion/teacher turnover is manageable to sustain the scheduling of physical activity.”

Polychoric correlation coefficients ranged from 0.28 to 0.87. Ten pairs of items recorded polychoric correlations above 0.8, reflecting possible redundancies. Of these 10 items, two were deemed appropriate to remove as they were considered adequately reflected by other items in the scale. The items removed included: “The scheduling of physical activity is well integrated into the operations of our school” from the Organisational Capacity domain, and “Evaluation results of the scheduling of physical activity are used to demonstrate success to funders and other key stakeholders (e.g., P&C, wider school community, etc.)” from the Program Evaluation domain. This resulted in 28 items being included in the psychometric evaluation.

### 3.4. Structural Validity

The initial model assuming no correlation between factors was a poor fit to the data across all fit indices and was subsequently improved in the first revised model by allowing factors to be correlated (see Table 3). The first revised model met the pre-specified criteria for adequate model fit according to the SRMR fit index but no others. One item from the Organisational Capacity domain was removed based on modification indices and review due to cross-loading with the Program Evaluation domain (“My school has a system for training new school champions/teachers to schedule PA”) (see Appendix A). The second revised model illustrated a slightly better fit to the data, illustrated by the improved fit indices and lower Akaike Information Criteria (AIC) (see Table 3). However, again only the SRMR index met the pre-specified criteria. A third revised model was calculated, further removing another item from the Organisational Capacity domain (“The level of school champion/teacher turnover is manageable to sustain the scheduling of PA”) due to correlation with other items from the Strategic Planning scale (see Table 3). All remaining items were considered theoretically important, and no further amendments were made to the model. The final items with their factor loadings are presented in Table 2.

### 3.5. Convergent Validity via Hypothesis Testing

Evidence for convergent validity tested via hypothesis testing was lacking, with small, non-significant associations recorded for all seven adapted PSAT domains and the total adapted PSAT score with teachers’ scheduling of weekly minutes of physical activity (Table 4).

### 3.6. Internal Reliability

All domain Cronbach’s alpha values were between the pre-specified threshold of 0.70 and 0.95, ranging from 0.77 to 0.92, and the total score was 0.95 (see Table 4).

### 3.7. Interrater Reliability

ICC(1) values ranged between 0.10 and 0.37 for the PSAT domains, indicating that between 10% to 37% of the total variance in adapted PSAT domain scores was attributable to differences between schools. The ICC(1) for the total adapted PSAT scores was 0.30 (Table 4). ICC(2) values ranged from 0.39 to 0.75 for the adapted PSAT domains, with only one domain, the Funding Stability domain, exceeding the 0.70 criteria for acceptable group-level reliability. The ICC(2) value for the total adapted PSAT score was 0.69 (Table 4).

### 3.8. Floor and Ceiling Effects

Fewer than 15% of respondents obtained the lowest or highest possible score for each of the seven domains. This indicates limited floor and ceiling effects of the measure (Table 4).

### 3.9. Norms

Domain scores ranged from a mean of 4.46 (SD = 1.12) to 5.30 (SD = 0.93), and a median 4.33 (Q1 = 4.00, Q3 = 5.33) to 5.40 (Q1 = 4.80, Q3 = 6.00), out of a possible range of one to seven (see Table 4).

## 4. Discussion

This study aimed to adapt and assess the validity and reliability of the PSAT as a measure for assessing the capacity for sustainment of teacher delivery of a physical activity policy in elementary schools. The final adapted 26-item PSAT for the elementary school setting illustrated adequate internal reliability for all domains, with Cronbach’s alpha values meeting acceptable thresholds. This finding is consistent with the reliability findings for the original PSAT [26]. However, the criteria for interrater reliability were only met for one domain, the Funding Stability domain, suggesting limited consistency in ratings from teachers from the same school for all other domains. This suggests that different individuals from the same school may have different views on the factors impacting on a school’s sustainment of a physical activity policy. Findings relating to indicators of validity were mixed, with inconclusive evidence for structural validity, no evidence of convergent validity from hypothesis testing, and no evidence of floor and ceiling effects, which suggests the potential responsiveness of the measure. The lack of strong evidence for validity suggests it is possible that we are failing to adequately capture the determinants of capacity for program sustainment in a school-based setting, at least in relation to the sustainment of scheduling physical activity across the school week.

Informed by stakeholders, implementation and school expertise, the original PSAT underwent extensive adaptation to the items and domains in order to ensure its relevance and appropriateness to the elementary school setting. This resulted in the entire “Partnerships” domain being removed as it focuses on issues of cultivating connections between the target program and stakeholders [26], an issue that is more relevant to a school’s governing body (e.g., educational department or schoolboard) than the individual schools, particularly teachers themselves. A further 11 individual items across six of the remaining seven domains were removed, also due to perceived lack of applicability to the individual school level. An additional six items were created: two to the “Funding Stability” domain and four to the “Organisational Capacity” domain. These new items were included as they were identified by the expert panel as important to the sustainment of health programs in schools (see Appendix A for the full list of amendments made). Furthermore, we found that most of the items had poor coverage of the lower end of the constructs being measured, with little variation and response at the lower end of the scale. These findings suggest that the adapted PSAT may not adequately measure the entire set of underlying constructs in this context and may need further refinement and specification to ensure it adequately captures both low and high levels of the domains. This is consistent with some previous studies that have used the PSAT and have reported domain scores skewed towards the higher end of the scale [44,45]. However, other studies have reported greater variation across domain scores of the PSAT [41,46,47,48].

In addition, no statistically significant relationship between the seven adapted PSAT domains or the total adapted PSAT score and teachers’ scheduling of physical activity were found. This is of concern, as sustained teacher scheduling of physical activity was the focus and primary outcome of the public health program that our implementation intervention was aiming to continue. Despite our intervention illustrating a significant and maintained effect on this outcome [32], the adapted PSAT failed to differentiate between teachers from the intervention schools with high and low scheduling behaviour, which we propose theoretically should reflect program sustainment. This finding highlights the potential usefulness of the adapted PSAT in its current form as a measure of determinants of sustainment of physical activity programs in elementary schools. However, while we did expect that higher scheduling of physical activity should reflect sustainment, this is only a proxy measure of sustained delivery of the policy and may not be truly reflective of this outcome. Given the multi-dimensional nature of sustainment, future research should ideally measure multiple indicators of sustainment, not just one. Furthermore, evidence of structural validity was limited, with only one of the fit indices from the CFA meeting the pre-specified criteria. Again, this indicates potential limitations in the validity of the adapted PSAT measure. However, most factor loadings were high (>0.40) and the findings from our CFA were only slightly poorer than those obtained from the psychometric evaluation of the original PSAT, where similarly only the SRMR met the criteria we used to determine adequate model fit [26]. Future studies with larger sample sizes and improved CFA methods are required to explore, in greater detail, the complex structure of this measure in order to gain a more robust understanding of the structural validity of the adapted PSAT.

The interrater reliability was only acceptable for the Funding Stability domain, highlighting the lack of consistency in individual teachers’ views from the same school on the determinants of capacity for the sustainability of a physical activity policy. It is possible that the specific factors being assessed by the adapted PSAT were not all relevant or appropriate for teachers to answer. While classroom teachers are frontline implementers who are instrumental to the day-to-day delivery of school-based public health programs, they often do not have authority over the organisational and external factors that are a large focus of the PSAT items. While a number of items not relevant to the school setting, or more specifically teachers, were removed to ensure relevance of the adapted PSAT, many of the remaining items are still possibly difficult for teachers to reliably respond to, such as allocation of funding and engagement of external providers to support implementation. Consequently, correct responses to some of the adapted PSAT items may require in-depth knowledge of schools’ organisational practices, which principals may decide but individual teachers are unlikely to be aware of. It has been recently argued that many existing measures of sustainment determinants may not be suitable for frontline individuals or practitioners [15], which is possibly the case in this instance. Sustainment is a complex, multidimensional process that is influenced by interactions across a range of multilevel determinants [13]. To completely understand the full range of factors that impact on program sustainment, different information may need to be collected from multiple sources and types of end-users and stakeholders. While it is encouraged that a range of stakeholders complete the PSAT [49], in the context of the school environment it may be important to develop separate measures that cover determinants of sustainment that are relevant to teachers and to school leadership or administrators. Future work is needed to determine the best approach to obtaining a comprehensive and accurate understanding of the determinants of sustainment.

### Limitations

There are several limitations that must be acknowledged when interpreting the current study results. First, despite involving members of the target population and experts in the content area to assist in the adaptation of the PSAT, we employed a relatively informal process of adaptation with experts who were involved in both the refinement of the scale and assessment of the face and content validity. If time and resources permit, future measure development studies should strive to include members of the target population in the development of the measure, while conducting cognitive interviews with a separate sample of the target population to ensure face and content validity [29]. Second, while we undertook methods to address the clustered nature of our data [38,39] and used the most conservative model, the method used assumes that the factor structure is the same at the individual and school level [38,39], which may not be the case. Unfortunately, we did not have an adequate sample or fit to explore the possible multilevel structure of the measure and were also limited in the modelling methods we could employ. Furthermore, no hypotheses concerning the multilevel structure of the adapted PSAT have been proposed, further limiting the ability to explore the higher-order factor structure. Larger samples with larger clusters are needed to conduct more robust, multilevel CFAs on the adapted PSAT. Third, the adapted PSAT for use in elementary schools was only validated in relation to one program, the scheduling of weekly classroom physical activity. Further research is needed to ensure it can be applied to assess sustainment of other public health programs in this setting. Fourth, there were slight differences in the implementation support strategies used across the two trials, which may introduce minor contextual differences that may impact on the measurement properties of the adapted PSAT. However, as the adapted PSAT was not measuring any aspect of the implementation strategies, such impacts should be minimal, if any. Finally, the adapted PSAT was designed to be completed by stakeholders with intimate knowledge of the organisational structures and factors, which may make it difficult for frontline staff to confidently and accurately complete all items in this measure [15]. Separate but complementary measures for administrative/leadership and frontline staff may be needed to ensure a comprehensive assessment of sustainment determinants is achieved.

## 5. Conclusions

If we are to ensure the long-term delivery and continued benefits of public health programs in schools, a clear understanding of the determinants of sustainment for school-based public health programs are needed, which is reliant on reliable, valid and pragmatic measures. Current attempts to adapt the PSAT for the elementary school setting among classroom teachers have been mixed, with evidence of internal reliability but mixed evidence of interrater reliability and validity. Future efforts to develop measures that are relevant and psychometrically robust for the elementary school setting and specific to the multiple end-user groups responsible for the delivery and governance of school-based public health programs are needed.

## Figures and Tables

**Table 1 ijerph-18-11414-t001:** School and teacher characteristics.

Characteristics	*n* (%)
**School level *n* = 30**	
**Sector**	
Catholic	14 (47%)
Government	16 (53%)
**Region**	
Inner/outer regional	9 (30%)
Major city	21 (70%)
**Trial**	
Pilot	5 (15%)
Effectiveness	25 (85%)
**Number of teachers (mean (SD))**	7 (5)
**Teacher level *n* = 214**	
**Gender**	
Male	33 (15%)
Female	181 (85%)
**Employment status**	
Permanent full-time	127 (59%)
Temporary full-time	59 (28%)
Permanent part-time	20 (9.4%)
Temporary part-time	8 (3.7%)
**Job share ***	
Yes	42 (20%)
No	168 (80%)
**Trial**	
Pilot	31 (16%)
Effectiveness	183 (86%)
**Age**—mean (SD)	40.7 (11)
**Years of teaching**—mean (SD)	15.2 (11)

* Frequency and percentages may not equal total sample due to missing values.

**Table 2 ijerph-18-11414-t002:** Item-level information for the final adapted PSAT.

Domain and Items	Missing N (%)	Standardised Factor Loading (se)	*p*-Value
**Domain: Strategic planning**			
My school has a sustainability plan (e.g., to continue the scheduling of the recommended minutes of physical activity long-term).	10 (4.7)	0.74 (0.04)	<0.001
My school’s goals to maintain the scheduling of PA are understood by all stakeholders (e.g., teachers, school champions, principals).	8 (3.74)	0.92 (0.02)	<0.001
My school clearly outlines roles and responsibilities to schedule PA for all stakeholders (e.g., teachers, school champions, principals).	8 (3.74)	0.91 (0.02)	<0.001
**Domain: Environmental support**			
There are champions within the school advocating for the scheduling of PA (a champion is someone who supports and advocates the policy, this may be your school executive or a teacher within the school).	2 (0.93)	0.70 (0.05)	<0.001
There are champions within the school with the ability to get resources for the scheduling of PA.	2 (0.93)	0.74 (0.04)	<0.001
My school has support from within the broader organisation i.e., DoE/CSO for the scheduling of PA.	4 (1.87)	0.71 (0.04)	<0.001
My school has support from outside our education department/office to help the scheduling of PA.	5 (2.34)	0.63 (0.05)	<0.001
The scheduling of PA for students at my school has strong public and community support.	5 (2.34)	0.68 (0.05)	<0.001
**Domain: Program adaptation**			
My school adapts or changes the scheduling of physical activity each week as needed (e.g., if PE equipment is damaged and cannot be used, heat wave etc).	2 (0.93)	0.74 (0.04)	<0.001
My school has a process to proactively adapt the scheduling of PA to meet changes in needs of the school community (e.g., to include other school programs).	2 (0.93)	0.82 (0.04)	<0.001
My school makes decisions about which physical activity components are ineffective and should not continue when scheduling PA (e.g., energizers, GoNoodle, running etc.)	3 (1.40)	0.59 (0.06)	<0.001
**Domain: Organisational capacity**			
School systems (e.g., space, time allocation) are in place to support the scheduling of PA.	1 (0.47)	0.71 (0.04)	<0.001
There are adequate resources and infrastructure within the school to schedule PA.	1 (0.47)	0.69 (0.04)	<0.001
School executives manage staff and other resources effectively to ensure that the scheduling of PA is met.	1 (0.47)	0.83 (0.03)	<0.001
My school has enough trained school champions to support the scheduling of PA.	1 (0.47)	0.74 (0.04)	<0.001
School champions and teachers at my school have enough supervision and support to implement the scheduling PA.	1 (0.47)	0.72 (0.04)	<0.001
*Note: The item “the scheduling of PA is well integrated into the operations of our school” was removed during the item assessment process.* *Items “The level of school champion/teacher turnover is manageable to sustain the scheduling of PA” and “My school has a system for training new school champions/teachers to schedule PA” were removed based on modification indices from CFA.*			
**Domain: Communications**			
My school has communication strategies in place to secure and maintain our school communities’ support for scheduling PA.	2 (0.93)	0.75 (0.04)	<0.001
Staff members at my school communicate the need for scheduling PA to the community (e.g., parents)	3 (1.40)	0.84 (0.03)	<0.001
My schools’ scheduling of PA increases community awareness of the need for PA in children	4 (1.87)	0.83 (0.03)	<0.001
**Domain: Program evaluation**			
My school has a system in place to actively evaluate the scheduling of PA (e.g., Improvements in children’s PA, student on-task behaviour etc.)	1 (0.47)	0.86 (0.03)	<0.001
My school reports the outcomes of scheduling the recommended minutes of PA (e.g., Improvement in student physical activity levels)	1 (0.47)	0.86 (0.03)	<0.001
Evaluation results inform the planning and implementation of the scheduling of PA.	1 (0.47)	0.81 (0.03)	<0.001
*Note: The item “Evaluation results of the scheduling of PA are used to demonstrate success to funders and other key stakeholders (e.g. P&C, wider school community, etc.)” was removed during the item assessment process.*			
**Domain: Funding stability**			
The school takes action to ensure there are ongoing funds to support the scheduling of PA. (e.g., included in annual school budget, funding from P&C)	2 (0.93)	0.75 (0.04)	<0.001
My school has a process in place to allow staff to attend professional development on scheduling PA (i.e., funding for ongoing professional development)	4 (1.87)	0.75 (0.04)	<0.001
My school provides time at work for staff to plan their schedule for meeting the recommended minutes of PA.	3 (1.40)	0.64 (0.05)	<0.001
My school can access a variety of funding sources to help schedule PA.	2 (0.93)	0.81 (0.04)	<0.001

**Table 3 ijerph-18-11414-t003:** Model fit statistics from confirmatory factor analysis (*n* = 214).

Model	SRMR	CFI	RMSEA	X^2^ (df), *p*-Value	AIC
Initial model—assuming there is no correlation between factors	0.319	0.644	0.135 (0.128, 0.142)	1517.64 (350), *p* < 0.001	13,259.10
Revised model 1—allowing factors to be correlated	0.073 *	0.803	0.103 (0.096, 0.111)	974.41 (329), *p* < 0.001	12,757.87
Revised Model 2—removing item from Organisational Capacity domain	0.070 *	0.820	0.100 (0.093, 0.108)	864.31 (303), *p* < 0.001	12,248.90
Revised model 3—removing item from Organisational Capacity domain	0.070 *	0.823	0.101 (0.093, 0.110)	804.08 (278), *p* < 0.001	11,844.87

* Indices meet the specified accepted criteria.

**Table 4 ijerph-18-11414-t004:** Domain-level results assessing the internal reliability, floor and ceiling effects, norms, hypothesis testing and interrater reliability.

Domain	Standardised Alpha	Floor *n* (%) ^a^	Ceiling *n* (%)	Mean (SD)	Median (Q1, Q3)	Minimum and Maximum Score	Hypothesis Testing Coefficient, *p*-Value *	ICC1	ICC2
Strategic planning	0.90	2 (1.0%)	12 (5.9%)	5.05 (1.15)	5.00 (4.33, 6.00)	1.0 and 7.0	−0.06, *p* = 0.981	0.21	0.60
Environmental support	0.84	0	5 (2.4%)	5.12 (0.91)	5.20 (4.60, 5.80)	2.0 and 7.0	1.97, *p* = 0.525	0.22	0.61
Program adaptation	0.77	0	10 (4.7%)	5.17 (1.02)	5.33 (4.67, 6.00)	2.0 and 7.0	2.47, *p* = 0.329	0.10	0.39
Organisational capacity	0.87	0	9 (4.2%)	5.30 (0.93)	5.40 (4.80, 6.00)	2.4 and 7.0	−0.04, *p* = 0.99	0.21	0.59
Communications	0.89	1 (0.5%)	7 (3.3%)	4.79 (1.03)	4.67 (4.00, 5.50)	1.0 and 7.0	0.08, *p* = 0.978	0.21	0.60
Program evaluation	0.92	1 (0.5%)	3 (1.4%)	4.46 (1.12)	4.33 (4.00, 5.33)	1.0 and 7.0	−2.21, *p* = 0.376	0.26	0.65
Funding stability	0.86	1 (0.5%)	7 (3.3%)	4.85 (0.98)	4.88 (4.25, 5.50)	1.0 and 7.0	−1.26, *p* = 0.667	0.37	0.75
Total PSAT	0.95	0	3 (1.4%)	5.00 (0.80)	4.98 (4.46, 5.54)	2.2 and 7.0	0.39, *p* = 0.914	0.30	0.69

^a^ Percentages may not correspond to total sample due to missing data. * Only those with valid physical activity scheduling data were included in the analysis.

## Data Availability

Data and materials used in this study are available from the study team on reasonable request.

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
