# Peer review of "Adaptation and Validation of the Program Sustainability Assessment Tool (PSAT) for Use in the Elementary School Setting"

_ijerph, 2021, doi:10.3390/ijerph182111414_

Round 1

Reviewer 1 Report

“2.6.2. Hypothesis testing” provides a hypothesis about the relationship between the PSAT score and teacher's total minutes of scheduled physical activity. However, the reason stated is "as this was the behavior in line with the policy that we were attempting to sustain in the two intervention trials." It is necessary to carefully explain why such a hypothesis was made in the light of previous research.

Author Response

We would like to thank the reviewer for their helpful comments. Please find below our response to their queries.

Reviewer 1

  1. “2.6.2. Hypothesis testing” provides a hypothesis about the relationship between the PSAT score and teacher's total minutes of scheduled physical activity. However, the reason stated is "as this was the behavior in line with the policy that we were attempting to sustain in the two intervention trials." It is necessary to carefully explain why such a hypothesis was made in the light of previous research.

We have updated the justification for our hypothesis proposing a positive relationship between the PSAT scores and delivery of physical activity at school. We have now clearly specified that our hypothesis is based on the theoretical underpinnings of the PSAT and the construct it is designed to measure, as well as findings from a previous study which has found an qualitative relationship between some of the PSAT domains and sustained behaviour (see page 7, para 2).

Reviewer 2 Report

The aim of this study was to adapt and evaluate the Program Sustainability Assessment Tool (PSAT) for the use in the elementary school setting. Here my comments.

Lines 38-44

I agree with this statement but, in my opinion, it is very general. Please consider of describing more in deep the different physical interventions in school setting, including more ecological interventions link run or walking (please refer for instance PMID: 31618975; PMID: 30092012)

Lines 93-106

Please explicit why do  you think the PSAT a good tool for elementary school setting. It was developed for assessing chronic disease prevention programs. I think that the resources in term of environmental support, funding stability, partnerships, organizational capacity, program evaluation, program adaptation, communications, strategic planning should be different in elementary school setting. In this line you described the PSAT, but you did not link it with your aim.

To confirm this, your PSAT is an adaptation of the original questionary with 30 items and not 40.

Aim

I think it is general and less informative of your research outcomes. I suggest to improve this part of the manuscript.

Lines 125-135

You want to evaluate the reliability and validity of the adapted PSAT. Nevertheless, you assess this using two different physical interventions in school setting. I think that this is an important bias of your research. To evaluate this aspect, all the conditions should be identical. May different types of intervention affect the results?

What about Convergent validity?

Lines 137-149

I suggest the authors to better describe the socio-cultural context and where the research was carried out. I think that this topic is important in this context due to the aim of the study. Indeed, to my knowledge, a study validation should considerer these aspects in order to avoid possible bias in the study validity.

Table 1

Please correct 15.2 (11%) and 40.7 (11%).

Trial: the sum is not 100%.

Author Response

We would like to thank the reviewer for their helpful comments. Please find below our response to their queries.

Reviewer 2

  1. Lines 38-44 I agree with this statement but, in my opinion, it is very general. Please consider of describing more in deep the different physical interventions in school setting, including more ecological interventions link run or walking (please refer for instance PMID: 31618975; PMID: 30092012)”.

We have now included specific details of the school-based physical activity programs that have been found to improve student rates of physical activity of fitness. We have focused on those programs that are supported by review evidence, rather than individual studies, to ensure that the most evidence-based data is presented (see page 2, para 1).

  1. Lines 93-106 Please explicit why do you think the PSAT a good tool for elementary school setting. It was developed for assessing chronic disease prevention programs. I think that the resources in term of environmental support, funding stability, partnerships, organizational capacity, program evaluation, program adaptation, communications, strategic planning should be different in elementary school setting. In this line you described the PSAT, but you did not link it with your aim. To confirm this, your PSAT is an adaptation of the original questionary with 30 items and not 40.

While the original PSAT has been predominantly psychometrically evaluated in the context of chronic disease programs at the community and state level, the developers do promote the PSAT as a general measure of determinants of sustainability of public health programs across multiple settings, including education settings. The strong pragmatic properties, theoretical underpinnings and flexibility to adapt the PSAT make it particularly appealing as a measure for adaptation to ensure the relevance and appropriateness for this setting. These are the main reasons why the PSAT was selected for adaptation. We have now provided explicit and extensive justification as to why the PSAT was selected for adaptation for the elementary school setting (see page 3, para 3).

  1. “Aim. I think it is general and less informative of your research outcomes. I suggest to improve this part of the manuscript.”

We have amended our aims to make them more specific to the purpose of the paper. They now reflect and state the specific psychometric and measurement properties that we aimed to assess in our evaluation of the adapted PSAT (see page 4, para 1).

  1. “Lines 125-135 You want to evaluate the reliability and validity of the adapted PSAT. Nevertheless, you assess this using two different physical interventions in school setting. I think that this is an important bias of your research. To evaluate this aspect, all the conditions should be identical. May different types of intervention affect the results?”

Apologizes for the confusion. The physical activity intervention that we were attempting to implement and sustain, and thus measure using the PSAT, were exactly the same in both trials. This intervention was weekly scheduling of classroom physical activity. The main difference between the two trials was the implementation strategies used to support the scheduling of physical activity, which were amended and improved from the pilot to the implementation trial. As the PSAT was designed to capture determinants of sustaining the physical activity intervention (i.e. scheduling of weekly classroom physical activity) and was not measuring any aspect of the implementation strategies, the minor differences between the trials should not impact the validation of this scale. We have clarified that there were no differences between the physical activity interventions in the two trials (page 4, para 4). However, we have also highlighted in the limitations section that there were slight differences in the implementation support strategies used in the two trials, which may introduce minor contextual differences that may impact on the measurement properties of the PSAT. However, as the PSAT was not measuring any aspect of the implementation strategies such impacts should be minimal if at all any (see page 17, para 2).

  1. “What about Convergent validity?”

We have clarified in the methods section that convergent validity was evaluated via the hypothesis testing assessing the relationship between the PSAT scores and the scheduling of weekly classroom physical activity (page 7, para 2). 

  1. “Lines 137-149 I suggest the authors to better describe the socio-cultural context and where the research was carried out. I think that this topic is important in this context due to the aim of the study. Indeed, to my knowledge, a study validation should considerer these aspects in order to avoid possible bias in the study validity.”

We have now included an extensive description of the sociocultural context of the region in which this research was carried out (page 5, para 2).

  1. “Table 1 Please correct 15.2 (11%) and 40.7 (11%). Trial: the sum is not 100%”

Thank you for identifying this typographical error. These numbers represent the mean and standard deviation. The percentage signs have now been removed (see Table 1).

Round 2

Reviewer 2 Report

No more comments. The authors